# Genetic Factors Affect the Survival and Behaviors of Selected Bacteria during Antimicrobial Blue Light Treatment

**DOI:** 10.3390/ijms221910452

**Published:** 2021-09-28

**Authors:** Joshua Hadi, Shuyan Wu, Aswathi Soni, Amanda Gardner, Gale Brightwell

**Affiliations:** 1AgResearch Ltd., Hopkirk Research Institute, Cnr University Ave and Library Road, Massey University, Palmerston North 4442, New Zealand; Joshua.Hadi@agresearch.co.nz (J.H.); Shuyan.Wu@agresearch.co.nz (S.W.); Aswathi.Soni@agresearch.co.nz (A.S.); Amanda.Gardner@agresearch.co.nz (A.G.); 2New Zealand Food Safety Science and Research Centre, Massey University Manawatu (Turitea), Tennent Drive, Palmerston North 4474, New Zealand

**Keywords:** antimicrobial blue light, blue light photoreceptor, blue light-sensing chemoreceptor, SOS-dependent DNA repair, antimicrobial resistance

## Abstract

Antimicrobial resistance is a global, mounting and dynamic issue that poses an immediate threat to human, animal, and environmental health. Among the alternative antimicrobial treatments proposed to reduce the external use of antibiotics is electromagnetic radiation, such as blue light. The prevailing mechanistic model is that blue light can be absorbed by endogenous porphyrins within the bacterial cell, inducing the production of reactive oxygen species, which subsequently inflict oxidative damages upon different cellular components. Nevertheless, it is unclear whether other mechanisms are involved, particularly those that can affect the efficacy of antimicrobial blue light treatments. In this review, we summarize evidence of inherent factors that may confer protection to a selected group of bacteria against blue light-induced oxidative damages or modulate the physiological characteristics of the treated bacteria, such as virulence and motility. These include descriptions of three major photoreceptors in bacteria, chemoreceptors, SOS-dependent DNA repair and non-SOS protective mechanisms. Future directions are also provided to assist with research efforts to increase the efficacy of antimicrobial blue light and to minimize the development of blue light-tolerant phenotypes.

## 1. Introduction

According to the latest Global Antimicrobial Resistance and Use Surveillance System (GLASS) report by the World Health Organization, there were 3,106,602 laboratory-confirmed infections related to antimicrobial resistant pathogens in the year 2019, including reports on an increasing trend of resistance to last-resort antibiotics [1]. In particular, the report highlighted the roles of *Escherichia coli* resistant to third generation cephalosporins and methicillin-resistant *Staphylococcus aureus* (MRSA) in causing bloodstream infections. Inevitably, researchers have been prompted to find alternative strategies for inactivating these pathogens beyond the use of antibiotics. These strategies include the use of electromagnetic radiation, such as ultraviolet (UV) and visible light [2]. It is a well-established fact that UV, particularly UV-C (200–280 nm), possesses potent bactericidal properties through the induction of pyrimidine dimers upon its absorption by nucleic acids [3]. However, safety issues may arise from the use of UV as a bactericidal agent, for example, increased risk of skin cancer in humans related to constant exposure to low-intensity UV [4,5] or long-term injuries associated with accidental exposure to high-intensity UV [6]. 

In contrast, visible light is less harmful than UV [7]. Within the visible spectrum, blue light at the wavelength range of 400–450 nm has the highest bactericidal property associated with its absorbance by endogenous porphyrins, leading to cell death [2,8]. Porphyrins are intermediate species in the heme biosynthesis [9,10] and may be activated by blue light to produce reactive oxygen species (ROS), such as superoxide, hydrogen peroxide, hydroxyl radicals, and reactive singlet oxygen (^1^O_2_) [11]. These ROS can subsequently induce oxidative stress upon different bacterial cell constituents, especially genomic materials, cell membrane, and cell wall (Figure 1A). Prokaryotic genomes can be degraded by ROS during blue light treatments [12,13], particularly through the oxidation of guanine residues into 8-hydroxydeoxyguanosine [12]. Fatty acids within the bacterial cell membrane are susceptible to blue light-induced oxidation, commonly marked by the formation of malondialdehyde [14,15]. Breakages of bacterial cell wall have been reported in Gram-positive MRSA [16] and Gram-negative *Acinetobacter baumannii* [17], and the inactivation of *E. coli* lipopolysaccharide has also been observed in vitro [18]. 

Furthermore, the efficacy of antimicrobial blue light extends to non-planktonic bacteria. Fila et al. demonstrated blue light-mediated inhibition of the quorum sensing signaling systems in *Pseudomonas aeruginosa*, leading to the delayed formation of biofilm, and subsequently reduced pathogenicity in the animal model *Caenorhabditis elegans* [19]. Besides nucleic acids, proteins, and lipids [19,20,21], polysaccharide materials contained within the biofilm matrix could also be targeted by antimicrobial blue light, albeit only when an exogenous photosensitizer (i.e., photoactive compounds, such as porphyrins) is applied [22].

Antimicrobial blue light treatments have mostly been utilized in clinical and food settings [23,24]. It involves the exposure of bacteria in a given matrix to a blue light source (for example, light-emitting diodes) at a fixed light intensity (Watt/cm^2^) for a period of treatment time (second). The efficacy of antimicrobial blue light treatments is most commonly measured by the reduction in bacterial counts after exposure to a certain light dosage (Joule/cm^2^), defined as the multiplication product of light intensity and treatment time. Importantly, sensitivities to blue light vary across bacterial strains, i.e., light dosage required to inactivate different strains may vary considerably [25,26,27]. Thus, as with other antimicrobial treatments, antimicrobial blue light may be applied in an insufficient amount (i.e., sub-lethal), potentially resulting in the development of tolerance [28]. However, the underlying mechanism of this phenomenon remains elusive and this represents an important research gap that needs to be addressed to improve future designs of blue light-based antimicrobial treatments, particularly in deciding whether complementary treatments are necessary. 

In this review, we summarize several pathways affecting bacterial responses to blue light, including when the blue light is applied at different growth temperatures, sub-lethal dosage or against varying forms of bacteria (planktonic, biofilm or spore). Beginning with the description of genes involved in the production of endogenous porphyrins, we subsequently discuss the roles of photoreceptors, chemoreceptors, and DNA repair-related genes in eliciting protective pathways against oxidative stress or in modulating bacterial physiological characteristics under blue light illumination (Figure 1B). 

## 2. Porphyrin, Bacteria and Antimicrobial Blue Light

It is an established fact that various bacteria synthesize porphyrins within their cell, and that the type and levels of porphyrins may vary across species [23]. The two most relevant porphyrins are protoporphyrin IX and coproporphyrin (I and III), as their involvements in antimicrobial blue light treatments have been demonstrated in both Gram-positive and Gram-negative bacteria, such as *S. aureus*, *P. aeruginosa*, *A. baumannii*, and *Helicobacter pylori* [29,30,31,32]. However, there is also evidence of differing sensitivities to blue light between Gram-positive and Gram-negative bacteria, primarily due to the varying levels of coproporphyrin produced in the two bacterial types [29]. In bacteria, the core pathway of 5-carbon heme biosynthesis usually begins with a charged glutamyl-tRNA that undergoes a series of enzymatic conversion processes to form coproporphyrinogen III via the universal intermediate species δ-aminolevulinic acid (ALA), also known as 5-aminolevulinic acid (Figure 2). At this point, two possible branches emerge from the core pathway: (1) Gram-negative bacteria decarboxylate coproporphyrinogen III to protoporphyrinogen IX, oxidize protoporphyrinogen IX to protoporphyrin IX and then add metal to form protoheme; (2) Gram-positive bacteria oxidize coproporphyrinogen III to coproporphyrin, insert iron to make coproheme and oxidize coproheme to protoheme [9,10]. Alphaproteobacteria possess ALA synthase and may undergo heme biosynthesis via the non-canonical 4-carbon pathway, in which glutamyl-tRNA is replaced by a compound derived from the combination of succinyl CoA from the tricarboxylic acid cycle and glycine [9]. 

The rate-limiting compound in heme biosynthesis is ALA, which acts as the committed precursor that leads to the formation of the tetrapyrrole structure of porphyrins [9]. Expectedly, external addition of ALA to light treatments has been reported to increase the production of porphyrins in *S. aureus*, *P. aeruginosa*, and *E. coli* (strain K-12 and Ti05), and subsequently render these bacteria more susceptible to light-based antimicrobial treatments [34,35]. Downstream of ALA, various enzymes are involved in the conversion of one tetrapyrrole structure to another, leading to the formation of different photoactive porphyrins (Figure 2). In addition to their respective functions in the forward reaction of heme biosynthesis, some of these enzymes are involved in feedback regulations, as demonstrated in *E. coli*, where the overexpression of *hemD* and *hemF* resulted in the accumulation of ALA, whereas *hemB*, *hemG* and *hemH* had a negative effect on ALA production [33]. The inhibition of HemB by protoporphyrin IX also occurs in *E. coli* [33]. Furthermore, there are other regulatory mechanisms in heme biosynthesis, including *hemA*-related regulations in *E. coli*, *Salmonella enterica* subsp. *enterica* serovar Typhimurium, *Bacillus subtilis*, and *P. aeruginosa* or oxygen-dependent regulations of coproporphyrinogen oxidase genes (i.e., *hemF* or *hemN*) in *Pseudomonas* spp., *E. coli*, and *B. subtilis* [10]. These findings indicate the possibility of variations in porphyrin species present at any given point, which could also be influenced by growth conditions. 

Porphyrin production has been suggested to depend on the bacterial growth stage. In *Aggregatibacter actinomycetemcomitans* and *Pophyromonas gingivalis*, the amounts of coproporphyrin (I and III) and protoporphyrin IX are highest in the first few days of growth, although variations have been observed across different types of porphyrins and bacterial strains [36]. Growth media could also impact porphyrin production, for example, *P. gingivalis* colonies cultured on a medium containing blood was reported to have increased production of protoporphyrin IX, relative to colonies grown on a medium lacking blood [36]. Consistently, there are evidences of strain-specific and growth phase-dependent inactivation of MRSA and *E. coli* by blue light, possibly related to porphyrin production [37,38]. In addition, a study revealed a positive correlation between ALA-mediated production of porphyrin species (protoporphyrin IX, coproporphyrin III, and uroporphyrin III) and elevated growth temperatures (up to 42 °C) in *Propionibacterium acnes* [39]. Nevertheless, there is still a need for further characterizations of endogenous porphyrins in other bacteria, along with assessments of their sensitivity to blue light, at different growth phases and temperatures. These data are particularly relevant for ensuring the appropriate design and application of blue light treatments, given that the relative absorption spectra of different porphyrin species vary. For example, protoporphyrin IX and uroporphyrin III absorb blue light at approximately 405–420 nm, whereas coproporphyrin III has the highest absorption at approximately 390 nm [40].

## 3. Heme Non-Producing Bacteria: Antimicrobial Blue Light or Photodynamic Therapy

Lactic acid bacteria (LAB), including pathogenic species belonging to the genera *Streptococcus* and *Enterococcus*, are not reliant upon heme for growth and thus unable to synthesize this cofactor, i.e., porphyrins are not prevalent in this group of bacteria [41]. Two studies have demonstrated that *Enterococcus faecalis* (formerly known as *Streptococcus faecalis* group D) and *Enterococcus faecium* devoid of porphyrins were insensitive to blue light-based antimicrobial treatments at 407–420 nm and 405 nm, respectively [29,42].

Intriguingly, others have reported on the inactivation of *E. faecium* by antimicrobial treatments with D_90_-values (light dosage required to obtain a 1-log inactivation) of 393 or 595 J/cm^2^ at 400 or 405 nm, respectively, which were considerably higher than other tested strains in both studies [25,43]. Insoluble fractions of the extracellular polymeric substance within *Streptococcus mutans* biofilms were also found to be susceptible to antimicrobial blue light (420 nm) [20]. However, the bactericidal mechanism of blue light against heme non-producing bacteria at these shorter wavelengths (400–420 nm) is yet unknown. In contrast, flavins are known to absorb blue light at longer wavelengths (450–460 nm), resulting in the inactivation of various bacterial species, including *E. faecium*, albeit with a lower efficacy than the conventional 405 nm antimicrobial blue light [43,44,45].

Alternative technology is also available in the form of antimicrobial photodynamic therapy (aPDT), which combines antimicrobial lights and photoactive compounds known as photosensitizers. Upon activation by light, photosensitizers at a ground state (lowest energy level) are converted into their excited singlet state (short-lived) or triplet state (long-lived), which, in the presence of oxygen, can undergo two types of energy transfer: (1) type I that produces toxic oxygen species, such as hydrogen peroxide (H_2_O_2_), superoxide or hydroxyl radicals; (2) type II that generates ^1^O_2_ [11]. Various photosensitizers and their derivatives are known to exhibit absorption within the ultraviolet/blue light spectrum [46], several of which have been applied in aPDT against various heme non-producing bacteria (Table 1). For a given aPDT, selection of a suitable photosensitizer and blue light combination, along with the optimization of photosensitizer concentration, is paramount to achieve maximum bactericidal efficacy [42,47,48,49,50].

## 4. Photoreceptors Facilitate Bacterial Responses to Blue Light

Light is ubiquitous in the environment and can be used as a regulatory signal by various organisms through photosensing. Photosensing is mainly mediated by six photoreceptors, namely rhodopsin, cryptochrome, BLUF (sensor of blue light using flavin adenine dinucleotide (FAD)), LOV (light, oxygen, and voltage), PYP (photoactive yellow protein), and phytochrome. In bacteria, phytochrome, BLUF, and LOV are the major photoreceptors [51,52]. Several bacteria may encode multiple photoreceptors, such as *Methylbacterium* spp. (phytochromes, LOV, and BLUF) or *Pseudomonas* spp. (phytochromes and LOV) [52]. As reviewed by Gomelsky and Hoff, these proteins are involved in controlling bacterial surface attachment, biofilm formation, and pathogenicity, among other functions [53]. Thus, future developments of blue light-based antimicrobial technologies could benefit from taking these photoreceptors into consideration, particularly their potential roles in conferring protection to bacteria against oxidative stress or in modulating the bacterial behaviors upon illumination by blue light at a sub-lethal dosage. In subsequent sections, we discuss the three major bacterial photoreceptors and their potential implications to blue light-based antimicrobial treatments. 

### 4.1. LOV-Mediated Response to Blue Light

LOV domains have been found across species of archaea, fungi, bacteria, protists, and plants [54]. A phylogenetic analysis has revealed that 115 of 3254 sequenced bacterial genomes (3.5%) carry at least one gene encoding the LOV domain, predominantly occurring in those belonging to the class Alphaproteobacteria or the phylum Cyanobacteria [55]. Among Firmicutes, LOV-encoding genes seem to appear exclusively in *Listeria* spp. and *Bacillus* spp. In 439 and 84 sequenced genomes of Enterobacteriales and Vibrionales, respectively, no LOV homologs were found, possibly due to the fact that these bacteria utilize other photoreceptor protein families, such as BLUF [55]. 

These LOV domains belong to the Per-Arnt-Sim (PAS) domain superfamily that contains a β-scaffold with five antiparallel β-strands (Aβ, Bβ, Gβ, Hβ, and Iβ) connected by four α-helices (Cα, Dα, Eα, and Fα) [56,57]. This core structure is usually flanked with helices at either the N-terminal (N-cap) or the C-terminal (C-cap) and these play a role in downstream signaling cascades [58,59]. First discovered in the plant *Arabidopsis thaliana* [60], LOV domains were subsequently reported to bind flavin mononucleotide (FMN) in a stoichiometric manner at their amino terminal, and thus may serve as a light sensor [61]. During light excitation, a conserved cysteine residue in the LOV domain active site forms a covalent adduct with the C4a atom of a flavin isoalloxazine ring (preferentially FMN, but also FAD) [57,62]. This induces a range of conformational effects, including the unfolding of flanking helices, dimerization, and rotation of the LOV modules [56,63]. 

The architectures of LOV domains across bacterial species are diverse (Table 2), primarily associated with different effectors, which are defined as the nearest conserved domain to the sensors with respect to the primary structure [54,63,64]. In bacteria, histidine kinase, sulfate transporter and anti-sigma factor antagonist (STAS), and GGDEF are among the known LOV effectors [54]. A phylogenetic analysis of 496 bacterial LOV domains has revealed that the majority of these existed in association with a histidine kinase (48%) or a GGDEF/EAL domain (21%), or as standalone short LOVs (14%) [65]. Interestingly, short LOVs appear almost exclusively in Proteobacteria, with the exception of the actinobacterium *Nakamurella multipartita* (strain Y-104) [66]. 

Downstream effects of different LOVs vary with their respective effectors [63]. For example, histidine kinases are a part of the bacterial two-component system that have the ability of autophosphorylation upon activation, facilitating subsequent transfer of phosphoryl group to response regulators to induce downstream effects, such as virulence [67,68]. As its name suggests, STAS acts as a transcription regulator through its activity upon RNA polymerase-binding sigma factors and is essential for bacterial responses to environmental stimuli, including during host colonization [69,70]. GGDEF is a domain involved in the synthesis or hydrolysis of cyclic-bis (3→5′) dimeric guanosine monophosphate (c-di-GMP) or in the hydrolysis of cyclic adenosine monophosphate (c-AMP), indicating its role in prokaryotic signal transduction [71].

In the current literature, there are only a few studies that have assessed the role of LOV domains in bacterial survival against blue light. Here, we focus on the role of LOV photoreceptors during blue light inactivation of *L. monocytogenes* and *B. subtilis*, as these two bacteria are well-described. Additionally, discussions are also provided for blue light-dependent behaviors of several bacteria with short LOV or LOV-HK domains and a novel RNA-binding LOV protein.

#### 4.1.1. Lmo0799 Photoreceptor and Transcription Factor Sig B (σ^B^) in *L. monocytogenes*

Numerous environmental stress responses in *L. monocytogenes* are regulated by a supramolecular complex called stressosome. These stressosomes consist of multiple copies of RsbR (or its paralogs) and RsbS proteins arranged into an icosahedron, with both proteins containing a STAS domain at their C-terminus. In association with the core structure, another protein called RsbT acts as a kinase to activate the stressosome under stress—phosphorylation of RsbS at the S59 residue or RsbR at the T205 residue during prolonged stress [72]. Post-phosphorylation of RsbS, the RsbT detaches from the stressosome and binds to the phosphatase RsbU, which in turn dephosphorylates RsbV to release RsbW. Subsequently, RsbW (an anti-sigma) is released from the transcription factor SigB (σ^B^) to allow downstream transcription of stress genes [72]. Indeed, σ^B^ is essential for mediating general stress responses in *L. monocytogenes* [73]. 

There are five RsbR homologs in *Listeria* spp., one of which is a protein called Lmo0799, which contains a LOV domain and acts as a photoreceptor [74]. Available evidences across various studies suggest that responses to blue light in *L. monocytogenes* is dependent on a functional Lmo0799, in association with σ^B^ [75,76,77,78]. In all of these studies, the motility of *L. monocytogenes* was reduced upon illumination by blue light, and this phenomenon has been attributed to the Lmo0799/σ^B^-dependent transcription of an antisense RNA [76]. In one transcriptomic study, an antisense RNA transcript was found to negatively regulate several motility genes in *L. monocytogenes* upon activation by σ^B^ [79]. This notion is further consolidated by the finding of motility repression by blue light in wild-type, but not in ∆*lmo0799* or ∆*sigB* strains [75]. 

O’Donoghue et al. explored the effects of blue light (460–470 nm) on the growth of *L. monocytogenes* (strains EGD-e and 10403S) as controlled by Lmo0799 and σ^B^ [77]. In the presence of sub-lethal blue light (i.e., lower light dosage), mutants that lacked σ^B^ exhibited a higher growth than the wild-type, suggesting an energetic cost of deploying σ^B^-controlled stress response. However, when the light dosage was lethal, a survival advantage was observed in bacteria carrying σ^B^ [77]. This finding was corroborated by Dorey et al., who showed that only ten genes were affected by blue light in *L. monocytogenes* mutants lacking functional σ^B^, as compared with 603 genes in the wild-type, confirming the importance of σ^B^ in blue light stress response [78]. Similar to the ∆*sigB* strains, *L. monocytogenes* mutants with deleted *lmo0799* gene or missense mutation at the position 56 of the *Lmo0799* protein (A56C mutation) were less affected by blue light at a sub-lethal dosage than the wild-type, as evidenced by the reduced motility repression and higher growth rate in these mutants than in the parental strains. Interestingly, when exposed to a lethal dosage of blue light, no difference in survival was observed between these mutants (∆*lmo0799* or missense mutation) and the wild-type, which is indicative of other σ^B^-activation pathways that bypass the canonical Lmo0799 light-mediated mechanism during severe stresses [77]. 

Furthermore, growth stage and temperature have also been reported to affect the survival of *L. monocytogenes* during blue light treatment, with bacterial cells in the stationary phase or cultured at 37 °C exhibiting less resistance against blue light than those in exponential phase or cultured at 30 °C, respectively [78]. These phenomena may be attributed to the varying levels of σ^B^ activation at different temperatures and growth stages, but not to the expression of *Lmo0799* photoreceptor that remained constant [78]. In support of this finding, another study has also reported that there is a reduced retention of the FMN chromophore by the *Lmo0799* photoreceptor at temperatures above 26 °C [80], possibly explaining the weaker light response of *L. monocytogenes* at higher temperatures. 

The relationship between blue light and virulence factors in *L. monocytogenes* also needs to be considered (Table 3). In one study, virulence factor ActA was essential for the activation of σ^B^ and biofilm formation under blue light illumination, suggesting its role in resisting oxidative stress [76]. Another study reported a more concerning observation, in which blue light at 455 nm induced the transcription of virulence genes *inlA* and *inlB* in *L. monocytogenes* (strain EGD-e), subsequently leading to an increased infection rate in Caco-2 enterocyte-like human cells, particularly when an additional salt stress was present [75]. In support of these findings, Prli42 protein in *L. monocytogenes* has been shown to enhance virulence factors (ActA and listeriolysin O) under oxidative stress induced by bone-marrow-derived macrophages (BMDMs) or hydrogen peroxide, with the wild-type strain having a higher survival against these BMDMs than the Δ*prli42* mutants [74]. Intriguingly, while it is only 31 amino acids in length (miniprotein), Prli42 may interact with RsbR and promote tolerance to oxidative stress induced by hydrogen peroxide [74]. 

#### 4.1.2. YtvA Photoreceptor and Transcription Factor σ^B^ in *Bacillus subtilis*

*B. subtilis* is a Gram positive, spore-forming bacterium, which has been used as a surrogate/model for understanding metabolism, resistance, and growth characteristics of other pathogenic spore formers. Similar to *L. monocytogenes*, *B. subtilis* has been reported to possess a signal integration hub acting as a stressosome. This stressosome is an icosahedral supramolecular complex consisting of RsbR (or its paralogs), RsbS and RsbT proteins, as determined by cryo-electron microscopy [87]. These proteins play various roles in signaling cascades involving Rsb proteins, including RsbT (kinase) and RsbU or RsbX (phosphatases), to activate σ^B^ [87,88]. An RsbR paralog, known as YtvA, is also present in *B. subtilis*—the first prokaryotic LOV protein to be identified—and it is capable of binding FMN and undergoing a photocycle [89]. Subsequent analyses have demonstrated the activation of YtvA by blue light through the cysteine 62 residue of its LOV domain, which induces a positive regulatory effect on the σ^B^-mediated stress responses [90,91]—it is noteworthy that YtvA was overproduced in both studies. A more recent study has shown that YtvA is permanently incorporated into the *B. subtilis* stressosome, regardless of whether light is present or absent [92]. 

As mentioned above, the blue light-induced activity of YtvA and σ^B^ in *B. subtilis* has only been observed using overexpressed YtvA, except when additional stress (for example, salt stress) is present, in which case wild-type expression of YtvA is sufficient to elicit signaling cascades to activate σ^B^ [93]. In addition, this activity is also reliant on the interaction between YtvA and RsbRA in the stressosome, possibly through their respective STAS domains [94,95]. Interestingly, the removal of an RsbR paralog called RsbRB also increased the blue light-induced activation of σ^B^ in *B. subtilis*, even when YtvA was expressed at wild-type levels [94]. These findings demonstrate the multi-factorial consideration that needs to be evaluated during an antimicrobial blue light treatment of *B. subtilis*, particularly treatment conditions (i.e., whether other stresses are present) and possible presence of individual strains with varying stressosome compositions. Furthermore, a research gap exists on the possible role of stressosomes in the initiation of sporulation as a response to light treatment. σ^B^ in *B. subtilis* has been reported to be involved in spore and biofilm formation, which are two important characteristics that help in survival against adverse conditions [96]. 

#### 4.1.3. LOV-Dependent Differential Physiological Behaviors of *Pseudomonas* spp. and *Rhodobacter sphaeroides* Containing Short LOV or LOV-Histidine Kinase

LOV-containing proteins are present across the genera *Pseudomonas*, such as in *Pseudomonas putida* (for example, PpSB1-LOV and PpSB2-LOV) and *Pseudomonas syringae* (for example, PST-LOV) [97]. The LOV proteins in *P. putida* consist of short LOVs, which are equipped with auxiliary structures on each end—located outside of the core LOV domain—known as the N-cap and C-cap (Jα-helix) [66,98,99]. These elements are essential for maintaining the structural integrity of *P. putida* photoreceptors, as a study has shown that the short and truncated variants of PpSB1-LOV and PpSB2-LOV had severe structural issues, which subsequently led to the loss of chromophore-binding capacity [66]. Distinctively, *P. syringae* contain LOV-HK domain in association with a response regulator (RR) [100]. This LOV-HK-RR construct in PST-LOV is essential for light-dependent regulation, primarily through the change in the interaction between the LOV-HK domain and the RR domain. However, the histidine kinase activity has also been reported to be conserved in the absence of RR domains, indicating two distinct mechanisms involved in the activity of PST-LOV, i.e., those related to the activation of histidine kinase alone or via interactions with RR domains [100]. 

The short LOVs of *P. putida* (strain KT2440) have been reported to be positive regulators of blue light-dependent transcription of genes controlled by the *P. putida* light-inducible transcriptional regulators (PplRs), which belong to the light-induced transcriptional regulator (LitR) family of proteins. These genes include *folE2*, *ufaM*, and *phrB* that encode a GTP cyclohydrase, a methylase involved in furan ring-containing fatty acid synthesis and a deoxyribopyrimidine photolyase, respectively. Albeit, the growth rate of *P. putida* was not affected under white-light illumination, as evidenced by the constant expression of the essential sigma factor *rpoD* [97]. Conversely, the growth of *P. syringae* (strain DC3000) is adversely affected by the presence of PST-LOV in a blue light-dependent manner [101]. The authors further demonstrated the direct involvement of PST-LOV in inhibiting the expression of several sigma factors and also in reducing the proliferation of *P. syringae* in *A. thaliana* under blue light illumination at 400–500 nm (Figure 3) [101]. 

Besides growth, blue light also affects the swarming motility of *P. syringae*, albeit in a strain-dependent manner. A group of researchers reported that blue light at 470 nm inhibited the swarming motility of *P. syringae* (strain DC3000), with this phenomenon partially attributable to the presence of PST-LOV photoreceptors [102]. However, another study found that while blue light induced the reduction in swarming motility (relative to dark conditions) in another *P. syringae* strain (strain B728a), the presence of LOV photoreceptors instead positively regulated swarming motility under illumination and this could be attributed to the histidine kinase activity of the LOV-HK domain within the PST-LOV [103]. Furthermore, the virulence of wild-type *P. syringae* (strain DC3000) is attenuated in the presence of blue light, as compared with strains lacking the LOV photoreceptors (Figure 3) [101,102]. 

Similar to the PpSB2 photoreceptor in *P. putida*, the LOV domain in the photosynthetic bacteria *R. sphaeroides* is not associated with any effector and equipped with extensions at the N-terminus (short α-helix motif) and C-terminus (helix-turn-helix motif) [104,105]. This LOV domain exists as a dimer under dark condition and turns into a monomer upon blue light illumination, suggesting its role in sensing light [105,106]. As demonstrated in one study, the LOV photoreceptor in *R. sphaeroides* (strain 2.4.1) primarily acted as a repressor in blue light-dependent gene regulations, particularly genes related to carbon metabolism and chemotaxis [107]. Furthermore, the authors revealed the role of *R. spaeroides* LOV protein in photooxidative stress, including that caused by blue light (450 nm). In particular, the lack of LOV protein resulted in the increased expression of RSP_4257, which is an anti-sigma-factor antagonist (STAS), upon blue light illumination, although the relationship between LOV photoreceptor and survival ability of *R. sphaeroides* against blue light is yet to be firmly established [107]. 

### 4.2. BLUF-Mediated Response to Blue Light

BLUF domains were first discovered through their photosynthetic functions in the purple bacterium *R. sphaeroides* [108,109] and their involvement in the photoavoidance response in the algae *Euglena gracilis* [110]. A BLUF domain consists of a five-stranded β-sheet and two ferrodoxin-like α-strands aligned parallel to the β-sheet [58,111]. The isoalloxazine ring of a bound FAD molecule is sandwiched between the α-helices, with its adenosine side-chain protruding to the protein surface [56]. This interaction with FAD is facilitated by conserved tyrosine, glutamine and methionine (or tryptophan) residues in the BLUF domain [111,112]. Indeed, in two models predicting the light-mediated conformational changes of BLUF, tyrosine and glutamine residues are essential elements, alongside the FAD active site [56,58,111]. BLUF is found in at least 10% of the sequenced bacterial genomes and the majority of these proteins possibly function within a two-component system—a photosensing system in which the effector occurs in other molecules, such as DNA or protein—and they are often called short BLUF proteins [52]. An example of this is the BlrB photoreceptor in *R. sphaeroides*, which is possibly involved in signal transduction by interacting with effector proteins, such as those containing the EAL domain, through the exposed amino acid residues located at the mouth of its flavin-binding moiety [113]. A range of other BLUF architectures has also been predicted in prokaryotic cells (Table 4) [114]. In this review, we focus on two bacterial BLUF proteins, namely YcgF in *E. coli* and BlsA in *A. baumannii*, as they are well-described in terms of their role in blue light perception and the concomitant downstream effects on the lifestyle of these bacteria.

#### 4.2.1. YcgF Photoreceptor in *E. coli*

Photosensing in *E. coli* is mediated by a photoreceptor known as YcgF. This protein contains a BLUF domain at its N-terminal and an EAL domain at the C-terminal. Interestingly, unlike other BLUF proteins, YcgF undergoes conformation change upon light illumination in a similar manner as LOV2 domain. Mediated by the Jα-helix at the C-terminal, this conformational change was initially thought to regulate the phosphodiesterase activity of EAL [115]. Unexpectedly, another study demonstrated that the EAL domain of YcgF lacked the phosphodiesterase activity to break down c-di-GMP [116], even when the missing conserved residues required for such activity were restored [117]. Instead, YcgF exhibited a regulatory role by antagonizing the activity of the YcgE protein, which is a repressor of eight genes, especially those present in the *ycgZ-ymgABC* operon—this operon is known to regulate the production of curli fimbriae and colanic acid. Upon illumination by blue light (470 nm), YcgF directly interacted with YcgE, resulting in the release of YcgE from its target operators, allowing for YmgA and YmgB to activate the Rcs phosphorelay pathway to induce the production of an exopolysaccharide component (colanic acid), promote acid resistance through the gene *bdm* and repress the expression of curli fimbriae [116]. This reaction is likely to occur through the blue light-induced conformation change in the EAL region of YcgF due to an altered hydrogen-bonding network around the FAD chromophore [118]. Furthermore, YcgF homologs have also been identified in *Klebsiella pneumoniae* and *Enterobacter* sp. 638, and a consistent pattern could be found in these bacterial species, in which a gene encoding a YcgF-like protein tends to occur next to a gene encoding a YcgE-related protein and a *ycgz-ymgAB*-like genetic unit [117]. 

Taken together, these findings indicate that YcgF, YcgE and the *ycgz-ymgAB* operon—or their homologs—constitute an important genetic pathway that controls blue light-dependent biofilm maturation and acid resistance in *E. coli* and potentially in other bacteria as well [116,117]. It is also noteworthy that the activity of YcgF upon YcgE is most prominent at low temperatures, particularly at 16 °C [116,118], albeit the downstream function of this pathway does not confer additional survival advantage against lethal blue light at this temperature [116].

#### 4.2.2. BlsA Photoreceptor in *A. baumannii*

*A. baumannii* is a Gram-negative anaerobic (obligate) pathogen that has been associated with nosocomial infections [119]. Photosensing in *A. baumannii* occurs through the BlsA photoreceptor. The X-ray crystal structure of BlsA has recently been solved and it is consistent with the canonical BLUF protein structure (as described above). The conserved residues in the flavin-binding moiety at the N-terminal are tyrosine (Y7), glutamine (Q51), and methionine (M94). Upon illumination, the α-helices at the C-terminal undergo conformation changes, which modify the affinity of effector proteins to BlsA, including BfmR (biofilm response regulator), Fur (iron uptake regulator), AraC-like regulators (virulence), and OmpR (osmotic stress response regulator) [120]. This is another example of a short BLUF photoreceptor, in which the signal transduction occurs through protein-protein interactions. 

Various studies have reported on the blue light-dependent regulation of motility, biofilm formation, and iron uptake in *A. baumannii*. One study demonstrated the inhibition of biofilm formation, pellicle formation, and motility in wild-type *A. baumannii* (strain ATCC 17978) by blue light (462 nm), whereas mutant strains lacking the functional *blsA* gene retained all three features [121]. Others unraveled the importance of the last fourteen residues at the C-terminal to the photodynamic properties of BlsA, as evidenced by the higher motility under blue light (469 nm) observed in strains with deleted (∆135–147) or substituted (K144E or K145E) residues than in the wild-type (*A. baumannii* ATCC 17978) [122]. The authors further suggested that Y7, Q51, and tryptophan (W92) residues of BlsA were also essential for the blue light perception of *A. baumannii* [122].

Metabolism of *A. baumannii* is also regulated through blue light. In iron deprived and dark conditions, BlsA binds to Fur, a repressor of acinetobactin, and promotes iron uptake. However, when activated by blue light (462 nm), BlsA detaches from Fur, which allows Fur to bind to its promoter, inhibiting iron uptake and growth [123]. In contrast, acetoin catabolism is induced by blue light (462 nm) through the interaction of activated BlsA with the repressor AcoN, allowing downstream expression of acetoin catabolic genes required for growth [124]. 

In all of these studies, blue light-mediated responses were only observed at ambient temperature (24 °C or lower), but not at elevated temperature (30 °C or above) [121,122,123,124]. This is expected as blue light photoactivity of BlsA is restricted to temperatures below 24 °C and that the expression of BlsA is mostly observed at 21–23 °C (especially in dark conditions), but negligible above 25 °C [125]. Furthermore, BlsA is unstable at high temperatures (above 30 °C) due to conformational changes that lead to the loss of FAD-binding capacity and protein aggregations, rendering the protein inoperative [125,126]. 

The virulence of *A. baumannii* is also affected by blue light, albeit the involvement of BlsA is yet to be determined. While blue light was found to increase the pathogenicity of *A. baumannii* in human keratinocyte cell line HaCaT at 37 °C, this phenomenon was independent of the BlsA photoreceptor [127]. In contrast, BlsA was reported to be involved in the killing of *tup1* mutant *Candida albicans* at 24 °C, as the authors highlighted the difference between the wild-type and the ∆*blsA A. baumannii* strains [121]. 

### 4.3. Blue Light-Sensing Bacteriophytochrome and PAS-Containing Photoreceptor in P. aeruginosa

Unlike several other Pseudomonas species, such as *P. syringae* and *P. putida*, *P. aeruginosa* does not encode LOV proteins in their genome [97]. Instead, *P. aeruginosa* senses light through the bacteriophytochrome PaBphP that utilizes biliverdin IXα (produced by the bacterial heme oxygenases) as its chromophore. As with other phytochromes, PaBphP functions through photointerconversion between the red light-absorbing (Pr) and the far red light-absorbing (Pfr) forms [128]. One study has found that BphP was also responsive to blue light (430 nm)—other than to red light (630 nm) and far red light (730 nm)—and subsequently mediated light-dependent inhibition of biofilm and virulence genes in *P. aeruginosa* (strain PA14), in association with the two-component system KinB-AlgB [129]. The authors further elaborated that, upon illumination, BphP activated the alginate biosynthesis transcriptional regulatory protein (AlgB) via phosphorylation to promote the aforementioned inhibition of biofilm and virulence genes. In contrast, Kinase B (KinB) inactivated AlgB through its phosphatase activity, which resulted in the promotion of biofilm and virulence [129]. Two other studies have shown similar results: (1) blue light (462 nm) impaired the replication ability of *P. aeruginosa* (strains PAO1 or PAE4840) in HaCaT [127]; (2) virulence factors, such as pyocyanin, hemolysins, elastases, proteases, and lipases, in various *P. aeruginosa* strains were inhibited by blue light at 405 nm [130]. 

A recent study demonstrated that the possible involvement of a photoreceptor known as Rmca in the blue light-dependent inhibition of biofilm maturation in *P. aeruginosa* (strain PA14). The authors showed that blue light (422 nm) had a profound inhibitory effect on the synthesis of cyclic di-guanosine monophosphate (c-di-GMP), which was later found to be the result of phosphodiesterase activities of four proteins, particularly Rmca [131]. As previously demonstrated, Rmca possessed GGDEF-EAL and PAS domains, with the latter containing residues capable of binding a flavin cofactor [132]. Nevertheless, the role of Rmca in direct sensing of light and in the regulation of biofilm development in *P. aeruginosa* remains to be fully elucidated. 

## 5. Role of Chemoreceptors in Responses to Blue Light

Five chemoreceptors (Aer, Tar, Tsr, Tap, and Trg) have also been reported to be involved in blue light-mediated tumbling (clockwise flagellar rotation) in *E. coli*. Wright et al. demonstrated that *E. coli* mutant strains expressing a single chemoreceptor (Aer or Tar) tumbled in response to blue light at 440 nm. A similar response was observed in strains expressing a combination of Aer and either Tar or Tsr, whereas the presence of Tsr alone failed to mediate any response to blue light [133]. More recently, Perlova et al. found that all five chemoreceptors responded to blue light (440 nm), with Aer, Tar, Tsr, and Trg initiating a tumbling response, whereas Tap-only strains exhibiting a running response under illumination. The authors also demonstrated that three functional proteins were required to produce blue light-mediated physiological changes, namely CheY (chemotactic signaling protein), CheB (methylesterase) and CheR (methyltransferase) [134]. As a well-established fact, chemotaxis in *E. coli* primarily relies on the histidine kinase CheA, which is coupled to the chemoreceptors through a linker protein (CheW). When the chemoreceptors are activated by a stimulus, CheA phosphorylates CheY, followed by the binding of the phosphorylated CheY to flagellar motors, causing a switch in the rotational direction of these motors—for example, counterclockwise to clockwise in a tumbling event [134,135]. CheB and CheR complete the signaling network by regulating the methylation of the chemoreceptors as a part of the sensory adaptation process (On or Off states) [134,135]. However, while Wright et al. and Perlova et al. have provided evidence of blue light-sensing capability of *E. coli* chemoreceptors, the precise mechanism is not fully elucidated. Aer contains a PAS domain that binds to FAD molecule, possibly explaining its response to blue light [136]. Nevertheless, the other chemoreceptors are not known to possess a chromophore-binding domain, and thus their blue light-related activities may be mediated by unknown blue light sensors elsewhere [133,134,135]. 

In contrast to *E. coli*, interlinked activities of blue light photoreceptors and chemoreceptors have been described in epiphytic *P. syringae* pv. tomato DC3000 (PsPto). Santamaría-Hernando et al. reported on the positive regulation of five PsPto methyl-accepting chemotaxis proteins (MCPs) by LOV and bacteriophytochrome (BphP1) photoreceptors under blue light illumination at 458 nm. More importantly, two of these MCPs (PSPTO_1008 and PSPTO_2526) are required for full virulence of PsPto [137], demonstrating the importance of blue light regulated pathways in the pathogenicity of plant pathogens.

## 6. Potential Development of Bacterial Tolerance to Blue Light via SOS-Dependent DNA Repair

In the event of oxidative stress, bacteria have various adaptive responses to enhance their survival. These adaptive responses could lead to the development of tolerance or resistance. To our knowledge of the current literature, no reports have identified full resistance of any bacterial species to antimicrobial blue light treatments. Nevertheless, one study has shown that five successive cycles of sub-lethal blue light (411 nm; 50 or 150 J/cm^2^ per cycle) resulted in the development of tolerance in *S. aureus* (strains USA300 JE2, community-associated methicillin-resistant; 1631, methicillin-sensitive; 2030, human-associated methicillin-resistant). More importantly, the authors found that this tolerance was stable after five successive culture transfers in the dark, and subsequently identified genetic alterations as the probable cause of the tolerance, particularly due to the expression of *recA* and *umuC* genes that triggered the SOS-dependent DNA repair [138]. Another study corroborated this result by demonstrating that sub-lethal exposures of *S. agalactiae* (strains ATCC 27956, 2306/07 or 2974/07) to blue/green light (515 nm; 10 cycles; up to 40 J/cm^2^ per cycle), in combination with a photosensitizer (rose bengal), altered the expression of six major oxidative stress response genes and led to the development of tolerance [139].

Mutations resulting from SOS-dependent pathway are best understood in *E. coli*, primarily through the translesion DNA synthesis (TLS), mediated by low-fidelity DNA polymerases, particularly pol V. Pol V consists of two UmuD homomers and one UmuC monomer (UmuD’2C) and is activated the RecA*, an assembly of recombinase A (RecA) protein associated with a single-stranded DNA. A RecA* nucleoprotein filament is known to activate pol V by transferring RecA-ATP complex from its DNA 3′-end to the UmuD’2C, obtaining the active pol V structure (UmuD’2C-RecA-ATP), also known as the pol V Mutasome (pol V Mut)—activation of pol V can only occur when RecA-ATP is transferred from the 3′-tip of RecA*, but not from the 5′-tip [140]. A region of the RecA protein, which encompasses amino acid residues 112, 113, and 117, has been identified as an important pol V activation surface [141]. During the induction of stress, the expression of genes in the SOS regulon is also initiated due to the release of the LexA repressor, as mediated by a RecA-dependent autocleavage. 

There is evidence implicating SOS-dependent pathways in the development of resistant phenotypes in several bacterial species (Table 5). In addition, SOS responses involve varying mechanistic modes across bacterial species. For example, SOS-inducing stresses have been reported to trigger the development of superinfectious phage Pf4 (and subsequently small-colony variants) or an integron rearrangement leading to the production of β-lactamases in *P. aeruginosa* [142,143]. In *S*. Typhimurium, SOS-dependent pathway is used as a strategy to avoid lethal antibiotic treatments by inhibiting swarming motility through the disruption of equilibrium between RecA and CheW proteins, preventing the bacterial chemoreceptor polar cluster assembly [144,145].

Furthermore, there are variations in the SOS regulon across bacterial species. The canonical pol V (*umuDC* operon) and *recA* are prevalent in *S. aureus*, *A. baumannii*, *S.* Typhimurium and *Vibrio* spp [146,147,148,149,150]. In contrast, *P. aeruginosa* lacks the *umuDC* operon and instead possesses the LexA-repressed *imuA/sulA-imuB-dnaE2* operon performing a similar function as the former [151]. Unlike other bacteria, *A. baumannii* does not have the LexA repressor, but instead possess an UmuD homolog, known as UmuDAb, which could undergo LexA-like cleavage in response to DNA damage and act as a RecA-dependent regulator of eight genes encoding seven error-prone DNA polymerases [152,153]. Taken together, these findings highlight the complexity of SOS-dependent stress responses existing in different bacteria, particularly their roles in promoting resistant bacterial phenotypes, and thus future studies are needed to explore their association with antimicrobial blue light-induced oxidative stresses.

## 7. Non-SOS Protective Mechanisms against Blue Light-Induced Oxidative Stress

Besides SOS-dependent pathways, several bacteria counteract blue light treatments by producing catalase, a ROS-scavenging enzyme. This enzyme is encoded by *katA* gene, which was found to be upregulated by 8-fold in *S. aureus* exposed to phototreatment at 462 nm and led to increased tolerance of the bacteria to hydrogen peroxide [156]. This is consistent with the findings of two other studies reporting increased tolerance to hydrogen peroxide in blue light-treated *S. aureus* [138,157]. Similarly, *P. aeruginosa* utilizes catalase to combat the photo-oxidative stress of blue light (464 nm), as evidenced by the higher susceptibility of mutants lacking *katA* to blue light than the wild-type [158]. Interestingly, one study demonstrated that blue light at 455 nm (225 J/cm^2^) was able to completely inactivate the enzymatic activity of a recombinant *P. aeruginosa* catalase A (KatA) in vitro, albeit the direct implication of the reduced KatA activity upon the blue light inactivation of *P. aeruginosa* remains unclear [159]. 

Furthermore, *S. aureus* produces staphyloxanthin, a membrane-bound carotenoid pigment with antioxidant properties. In a recent study, *S. aureus* possessing the *crtM* gene, which is involved in the production of staphyloxanthin, exhibited a tolerant trait against blue light at 405 nm [160]. Thus, the authors proposed a pre-treatment of the tolerant *S. aureus* with blue light at 460 nm—which did not exhibit direct bactericidal activities but was capable of promoting the degradation of staphyloxanthin—resulting in the increased efficacy of the subsequent treatment with 405-nm blue light. In support of this finding, another study showed that exposure to blue light at 460 nm degraded staphyloxanthin and rendered methicillin-resistant *S. aureus* susceptible to subsequent treatments with oxidative agents [161]. 

As described in one study, *Cronobacter sakazakii* responded to blue light by upregulating three genes encoding an oxidative stress-resistance chaperone (*ESA_RS13255*), an adhesin (*ESA_RS09025*), and a capsule biosynthesis protein (*ESA_RS15435*), with the former promoting higher antioxidant activities [15]. For *B. subtilis* spores, resistance to blue light (400 nm) depends on the presence of DNA protective α/β-type small, acid soluble proteins (SASP), DNA repair proteins (nucleotide or base excising repair proteins and spore photoproduct lyase), spore coat proteins and proteins for spore pigmentations—mutants lacking genes encoding these proteins were more susceptible to photoinactivation [162]. 

In *V. cholerae*, blue light-mediated oxidative stress response is controlled by an anti-sigma ChrR (represses SigmaE) and metalloregulatory-like (MerR) proteins, either together or separately. Both proteins control 222 genes that are responsible for a range of functions, including cellular protection, DNA repair, and carbon metabolism [163]. Intriguingly, while cryptochrome/photolyase proteins are the sole blue light photoreceptors in *V. cholerae* [164], these photoreceptors are not involved in the bacterial photo-oxidative responses, indicating possible involvement of pigments or unknown photoreceptors in the *V. cholerae* stress response to blue light [163]. That being said, Tardu et al. did not analyze specific DNA repair pathways in *V. cholerae*, for example, photoreactivation involving the photolyase protein, and thus this is a subject of future studies. 

## 8. Concluding Remarks and Future Outlook

The complex biological pathways involved in bacterial responses to antimicrobial blue light present challenges to be overcome. Endogenous porphyrin-mediated oxidative stress remains the primary mechanistic model of antimicrobial blue light. However, given the diversity of porphyrin species occurring at different stages of the heme biosynthesis, and thus corresponding to different blue light absorption spectra, it is paramount for future studies to assess the porphyrin biosynthesis pathway as a function of culture conditions, such as bacterial growth stage, growth media, and temperature. Subsequently, a correlation between bacterial growth stages, porphyrin species present and survival rate against antimicrobial blue light can be established. 

The lack of understanding of blue light-mediated inactivation of heme non-producing pathogenic bacteria is another research gap that needs addressing. These bacteria may import exogenous heme into the cell and assemble heme-containing proteins, such as cytochrome *bd* and catalase. A small set of heme-binding proteins has also been discovered in LAB, including alkyl hydroperoxide reductase C (AhpC) and HemW, which exhibit blue-light absorptions at 414 and 425 nm, respectively, when loaded with heme [41]. This could be explored further to devise a strategy to inactivate these bacteria in the presence of external heme. Furthermore, there are other uncharacterized endogenous photosensitizers that absorb blue light at longer wavelengths, especially at 470 nm, which may be targets for antimicrobial blue light treatments in the future [40].

Photoreceptors can confer protective effects through blue light-dependent activation of stress response pathways and promote virulence when the light dosage is sub-lethal. In addition, several bacteria encode more than one type of photoreceptor. This characteristic is particularly prevalent in plant pathogens, which could encode up to four types of photoreceptor, as demonstrated in *Xanthomonas citri* subsp. *citri* containing bacteriophytochrome, LOV, BLUF, and cryptochrome/photolyase [165]. Intriguingly, one study has demonstrated a blue light-dependent inhibitory activity of LOV photoreceptors against a bacteriophytochrome (BphP1), leading to the differentiated swarming motility of *P. syringae* (strain B728a) [103]. Similarly, the interaction of AppA (a BLUF-containing protein) and CryB (a cryptochrome) is also known to regulate photosynthetic genes of *R. sphaeroides* in a blue light-dependent manner [166]. Future studies may be aimed at revealing overlapping activities of different photoreceptors in other bacterial species, along with their downstream effects on the bacterial growth and/or physiological characteristics under blue light illumination. 

Some bacteria are able to use chemoreceptors to modulate their motility or virulence in a blue light-dependent manner. While the underlying mechanism is not fully understood, there is evidence of synergistic interaction between chemoreceptors and photoreceptors in PsPto. In *E. coli*, the Aer chemoreceptor contains PAS domain, possibly indicating a capability of direct light sensing [136]. The next research step in this area may include screening of other chromophore-binding domains in known chemoreceptors present across different bacterial species, particularly pathogenic strains. 

As with other antimicrobial treatments, antimicrobial blue light is prone to sub-lethal application, especially considering that bacteria possess protective mechanisms against oxidative stress, such as ROS-scavenging catalases. More concerningly, low-fidelity polymerases expressed during the SOS-dependent stress response could result in the emergence of tolerant phenotypes during blue light treatments, as demonstrated in *S. aureus* [138]. Thus, further investigations should be conducted to unravel the mechanisms and frequencies of blue light tolerance, or resistance, in other bacteria [28]. In addition, alternative strategies can be developed using compounds capable of inhibiting different components of the SOS pathway. In *S. aureus*, the inhibition of RecA by novobiocin, as mediated by a DNA gyrase subunit B (GyrB)-dependent mechanism, was reported to increase the susceptibility of the bacteria to photoinactivation by about 100-fold, including to blue light at 405 nm [13]. Similar effects were observed for a recombinant *E. coli* containing an α-helix peptide called 4E1—this peptide contains 20 amino acid residues that match the motif of RecX protein (a natural RecA inhibitor)—which was rendered more susceptible to UV than the wild-type [167]. There are other inhibitors targeting different SOS elements, such as LexA, RecBCD, pol IV, and single-strand DNA binding proteins [168]. Future studies can aim to explore the possible addition of these inhibitors into antimicrobial blue light treatments against different bacteria.

Taken together, the mechanisms presented here highlight several inherent characteristics that set one bacterial species apart from another. Consequently, blue light inactivation parameters must be validated within the context of specific bacterial strains. In addition to these endogenous factors, external treatment conditions, such as light intensity, light dosage, availability of nutrients, temperature, and presence of exogenous photosensitizers, are known to contribute to the varying blue light inactivation profiles in clinical and foodborne bacteria [23,24]. Thus, the frontier of research in this area needs to be extended to include systematic determination of both inherent and external factors affecting the efficacy of antimicrobial blue light against different bacterial strains, with the subsequent establishment of accessible databases for improved predictability of bacterial tolerances or resistances. 

## Figures and Tables

**Figure 1 ijms-22-10452-f001:**
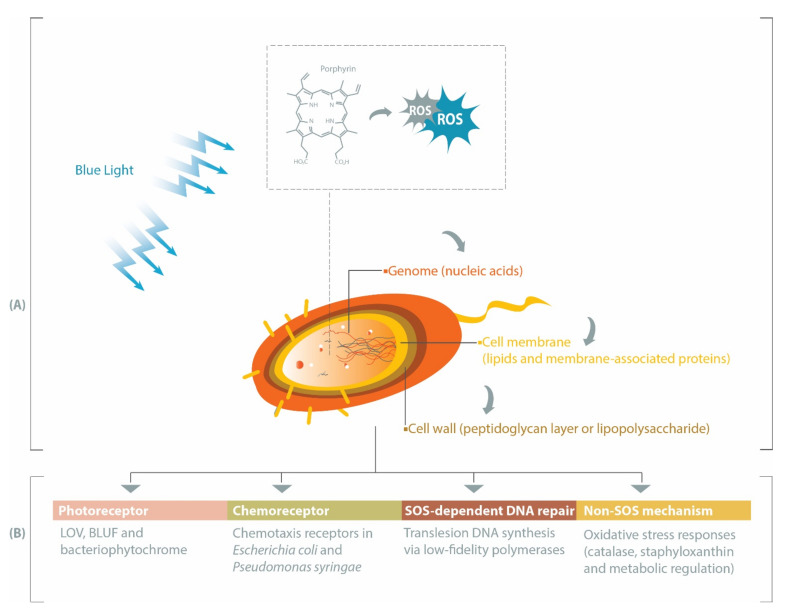
Pathways involved during blue light-mediated antimicrobial treatment. (**A**) Widely accepted inactivation mechanism of antimicrobial blue light, in which endogenous porphyrins are activated to form reactive oxygen species (ROS) that subsequently inflict oxidative stress upon different parts of the bacterial cell. (**B**) Bacteria can respond to blue light through four other pathways, leading to differential physiological behaviors and survival rates, which are discussed in this review.

**Figure 2 ijms-22-10452-f002:**
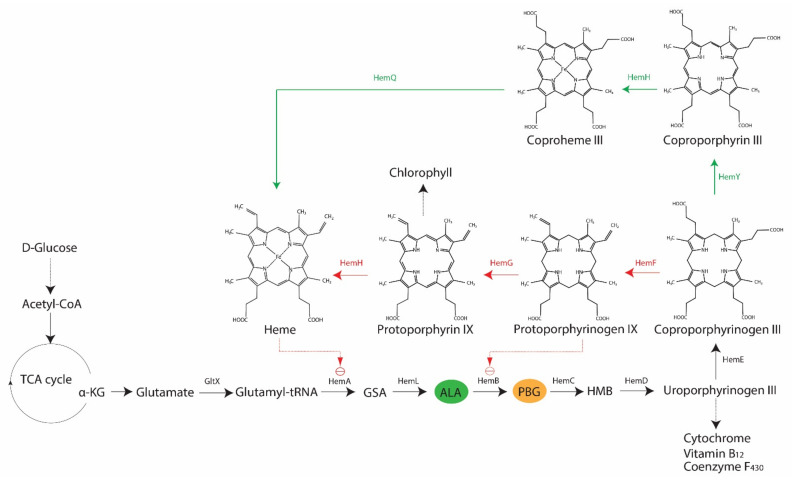
Biosynthesis of heme via 5-carbon pathway in bacteria. In the core pathway, glutamyl-tRNA is converted to coproporphyrinogen III, after which two branches emerge: Gram-negative (red arrows) and Gram-positive (green arrows) bacteria. Dotted red arrows signify feedback-inhibition. This figure was recreated and modified from the work by Zhang et al. (2015) [33], which was published under Creative Commons Attribution 4.0 International License (https://creativecommons.org/licenses/by/4.0/, accessed on 28 July 2021). TCA: tricarboxylic acid; α-KG: α-ketoglutarate; GltX: glutamyl-tRNA synthetase; HemA: glutamyl-tRNA reductase; HemL: glutamate-1-semialdehyde aminotransferase; GSA: glutamate-1-semialdehyde; ALA: 5-aminolevulinic acid; HemB: 5-aminolevulinic acid dehydratase; PBG: porphobilinogen; HemC: porphobilinogen deaminase; HMB: hydromethylbilane; HemD: uroporphyrinogen III synthase; HemE: uroporphyrinogen carboxylase; HemF: coproporphyrinogen III oxidase; HemG: protoporphyrin oxidase; HemH: ferrochelatase; HemY: coproporphyrinogen III oxidase; HemQ: iron-coproporphyrin decarboxylase.

**Figure 3 ijms-22-10452-f003:**
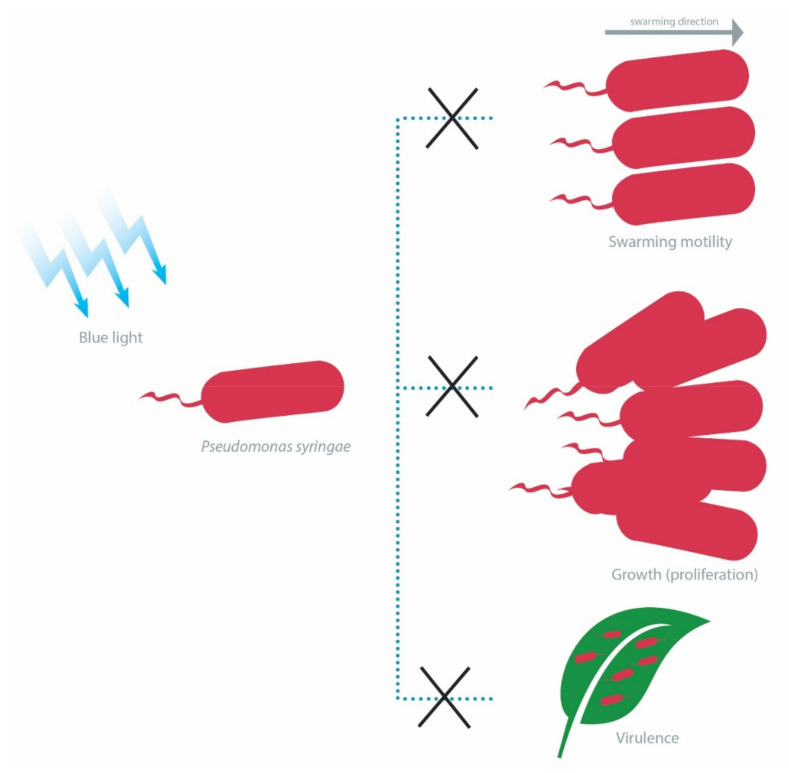
Blue light inhibits swarming motility, growth and virulence of *Pseudomonas syringae*.

**Table 1 ijms-22-10452-t001:** Selected in vitro studies on photodynamic treatment of heme non-producing bacterial species.

Bacterial Species	Blue Light	Photosensitizer ^a^	Light Dosage (Joule/cm^2^) *^, a^	Bactericidal Efficacy **^, a^	Reference
*Streptococcus agalactiae*	450 nm (pulsed)	CP III (0.08 mg/mL) PP IX (0.08 mg/mL)	7.6	8.89 log CFU/mL 9.54 log CFU/mL	[47]
*Streptococcus mutans*	360–550 nm	curcumin (2 µM)	54	2 log cells ***	[48]
*Streptococcus pneumoniae* (planktonic or biofilm)	405 nm	chlorin e6 (10 µM)	12 or 90	approximately 5 or 6.5 log CFU/mL	[49]
*Enterococcus faecalis* (planktonic or biofilm)	405–500 nm	eosin Y (5 or 10 µM) rose bengal (1 or 2 µM) curcumin (5 or 1 µM)	108	4.9 or 13.8 log CFU/mL 7.3 or 13.8 log CFU/mL 7.6 or 13.7 log CFU/mL	[50]
*Enterococcus faecium*	405 nm	curcumin (1 µg/mL) PP IX (0.1 µg/mL)	25.3	significant drop in optical density (655 nm) at *p* < 0.001	[42]

CP, coproporphyrin; PP, protoporphyrin; CFU, colony forming units. * Light dosage was calculated by multiplying light intensity (Watt/cm^2^) with treatment time (second). ** Bactericidal efficacy is defined as the reduction in the number of viable bacteria induced by antimicrobial blue light treatment. *** Number of bacterial cells was quantified by flow cytometry. ^a^ Photosensitizer concentrations, light dosages and bactericidal efficacies separated by the word “or” indicate differences between planktonic or biofilm cells.

**Table 2 ijms-22-10452-t002:** Selected prokaryotic proteins containing different LOV domains.

Bacterial Species (Phylum/Class)	NCBI Accession Number of Parent Protein	LOV Domain	Number of Amino Acids
*Listeria monocytogenes*, *Listeria innocua*, *Bacillus subtilis*, *Bacillus cereus* (Firmicutes)	WP_003738433.1 (LM), WP_003761135.1 (LI), WP_047183059.1 (BS) or AUZ27534.1 (BC)	LOV+STAS	253 (LM and LI) or 261 (BS and BC)
*Acinetobacter baumannii* 1294596 (ɣ-Proteobacteria)	EXF56192.1	GAF+PAS+LOV+GGDEAL+EAL	855
*Shewenella putrefaciens* (ɣ-Proteobacteria)	WP_025008327.1	MASE+CHASE+PAS+LOV+GGDEF+EAL	1216
*Arthrospira maxima* (Cyanobacteria)	WP_006668677.1	GAF+PAS+LOV+PAS+GAF+Kinase	1184
*Nakamurella multipartita* (Actinobacteria)	WP_015749472.1	PAS+RR+LOV	365
*Brucella abortus* (α-Proteobacteria)	Q2YKK7.2	LOV+PAS+HK	489

LOV, light, oxygen, and voltage; STAS, sulfate transporter and anti-sigma factor antagonist; GAF, cGMP-specific phosphodiesterases, adenylate cyclases and FhlA; PAS, Per Arnt Sim; GGDEAL, diguanylate cyclase named after conserved amino acids; EAL, diguanylate phosphodiesterase named after conserved amino acids; MASE, membrane-associated sensor 1; CHASE, cyclases/histidine kinases associated sensory extracellular; RR, CheY-type response regulator; HK, histidine kinase.

**Table 3 ijms-22-10452-t003:** Virulence factors in *Listeria monocytogenes* that are affected, directly or indirectly, by blue light treatments.

Virulent Factors	Description	Reference
ActA	ActA functions as a bacterial defense against autophagy and is controlled by the transcription factor PrfA. Inside the host cell cytoplasm, ActA recruits host cell cytoskeletal proteins to inhibit ubiquitin and p62, which renders *L. monocytogenes* unrecognizable during the autophagic process, allowing the bacteria to proliferate.	[81,82]
InlA and InlB	Invasins that are essential for internalization into the host cell. The efficacy of InlA and InlB varies across *L. monocytogenes* strains and types of host cell (particularly the expression of relevant receptors on different host cells). It is also known that InlA and InlB are regulated by σ^B^, with all three components needed for an effective infection of *L. monocytogenes*.	[83,84,85]
Listeriolysin O	Cholesterol-dependent cytolysin that primarily plays a critical role in breaking the membrane of phagosomes post-internalization, allowing *L. monocytogenes* to invade the cytosol. Pore-forming activity of listeriolysin O may also facilitate the internalization of *L. monocytogenes* at the early stage of infection in a calcium ion- and potassium ion-dependent manner.	[84,86]

ActA, actin assembly-inducing protein; InlA, internalin A; InlB, internalin B.

**Table 4 ijms-22-10452-t004:** Selected prokaryotic proteins containing different BLUF domains. Data presented were obtained from Kaushik et al. (2019), which was published under Creative Commons Attribution 4.0 International License (https://creativecommons.org/licenses/by/4.0/, accessed on 28 July 2021.) [114].

Bacterial Species	NCBI Accession Number of Parent Protein	BLUF Domain (Predicted Function)	Number of Amino Acids
*Escherichia coli*	ARH96915.1	BLUF+EAL (regulation of diguanylate cyclases and phosphodiesterase activity)	403
*Rhodobacter sphaeroides ATCC 17025*	ABP71929.1	BLUF+B12-binding (enhancement of photosensing capability)	448
*Leptonoma illini DSM 21528*	EHQ08139.1	BLUF+CHD (regulation of diguanylate cyclases)	309
*Methylobacterium radiotolerans*	WP_012321331.1	BLUF+PRK09039 superfamily (regulation of phosphoribulokinase, uridine kinase and panthothenate kinase activity)	309
*Hymenobacter* sp. *PAMC 26554*	AMR27912.1	BLUF+REC (regulation of chemotaxis)	292
*Curtobacterium luteum*	WP_058726129.1	BLUF+AcrR (regulation of antibiotic resistance)	335
*Thiocystis violascens DSM 198*	AFL74487.1	BLUF+EAL+GGDEF (regulation of c-di-GMP level)	597
*Legionella steelei*	WP_058511962.1	BLUF+PAS (regulation of cellular signaling)	402

BLUF, sensor of blue light using flavin adenine dinucleotide; EAL, diguanylate phosphodiesterase named after conserved amino acids; B12, vitamin B12; CHD, cyclase homology domain; PRK, phosphoribulosekinase; REC, cheY-homologous receiver domain; GGDEF, diguanylate cyclase named after conserved amino acids; c-di-GMP, cyclic di-guanosine monophosphate; PAS, Per-Arnst-Sim.

**Table 5 ijms-22-10452-t005:** Selected studies on sub-lethal SOS-inducing stresses resulting in differential bacterial phenotypes.

Bacterial Species	SOS Activator	Physiological Manifestation	Reference
*Staphylococcus aerus*	H_2_O_2_	formation of gentamicin-resistant and H_2_O_2_-tolerant SCVs due to enhanced catalase production mediated by pol V, RecA and RexAB proteins	[154]
*Acinetobacter baumannii*	UV, MMS (alkylation), ciprofloxacin and dessication	increased prevalence of rifampin-resistant mutants as mediated by RecA protein	[152]
*S. enterica* subsp. *enterica* serovar Typhimurium	MMC	decreased swarming motility to avoid lethal MMC, as controlled by RecA protein, but not other proteins involved in SOS response	[144]
*Pseudomonas aeruginosa*	H_2_O_2_, nitric oxide and MMC	formation of superinfective phage Pf4, resulting in the appearance of SCV within the bacterial biofilm	[142]
*Vibrio cholerae*	Antibiotics ^1^	increased prevalence of rifampin-resistant mutants	[155]

SCV, small-colony variants; UV, ultraviolet; MMS, methyl methanesulfonate; MMC, mitomycin C. ^1^ antibiotics used were ciprofloxacin, trimetophrim, ampicillin, aminoglycosides (spectinomycin, tobramycin or neomycin), chloramphenicol or tetracyclin.

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
