# Peer review of "Genetic Factors Affect the Survival and Behaviors of Selected Bacteria during Antimicrobial Blue Light Treatment"

_ijms, 2021, doi:10.3390/ijms221910452_

Round 1
Reviewer 1 Report
In this review, the authors described several pathways involved during blue light treatment. Four pathways photorecepter / Chemoreceptor .SOS and Non-SOS mechanism were summarized in this article. The author collects data very carefully and summarizes it well. However, the information is too much and messy. Therefore it is recommended to shorten the article. I hope that the author will make a summary figure in each paragraph, which will help readers understand the content clearly.
Reviewer 2 Report
In the review article “Genetic factors affect bacterial survival and behaviors during antimicrobial blue light treatment”, Hadi at all present information regarding the bacterial response to blue light treatment, an alternative therapy to traditional antibiotic drugs.
While the information provided is helpful to someone studying blue light therapy, a serious oversight is apparent in that the authors present the paper as though all bacteria are inherently susceptible to blue light therapy. What about organisms that do not encode porphyrins (those that do not synthesize heme), such as S. mutans, mentioned in Line 60? Can you hypothesize an alternative process, or at least mention that alternative pathways must exist? Most streptococci do not produce heme, but rather must import it. Therefore these organisms do not have some (or in some cases any) of the porphyrins required for biosynthesis of heme and thereby for efficacy of the blue light treatment. The authors make no mention of photodynamic therapy, in which a photosensitizer is provided in conjunction with the light treatment, as without it, the bacteria are not sensitive to the light. The omission of this information is a glaring oversight that must be corrected if this review is to be published.
As the authors focus their description on details of relevant enzymes of selected organisms such as L. monocytogenes, B. subtills, P. aeruginosa, E. coli, and A. baumannii, the title of the review should be changed to make clear that the information focuses on selected bacteria, as the current title is too broad.
My specific comments are as follows:
Line 44: change “relatively safer” to “more safe” or “less harmful”
Lines 65-67: The sentence beginning “Similar to the inactivation of planktonic cells…” is a bit awkward. I don’t think that the reader would expect that the mechanisms of inhibition of cells in a biofilm would necessarily be different than planktonic. Maybe just rephrase to indicate that the degradation of EPS is a further mechanism.
Line 76: Streptococcus faecalis group D is now referred to as “Enterococcus faecalis” (and has been for over 30 years)
Might be nice to do a careful search on non-porphyrin-producing organisms against which the blue light treatment has been tested in the lab. This would be a nice topic to include toward the end of the review.
Paragraph beginning p. 70 – consider moving this to the end when discussing future directions. It’s a bit distracting to bring up the downsides/uncertainties so early in the review.
Are there reports of blue light treatment being used in practice, outside of the lab, yet? This would be a good topic with which to replace this paragraph.
Figure 1: Overall, this figure is not very helpful. In part A, under the word “Photosensitizer”, include “(porphyrins)”. Adjust the illustration so it is clear that the photosensitizer naturally exists inside the cell and that the ROS are thereby generated inside the cell. Maybe make the image of the cell itself larger so that these portions can fit inside of it. Alternatively, a more helpful figure would be an illustration of the blue portion of the light spectrum, with indications of treatments that have been tested at various wavelengths. This would help to give context to the reader when these wavelengths are listed throughout the review.
Lines 211-214: These lines mentioning how N. multipartita may have acquired a short LOV are unnecessary for this review. At best, a guess is offered, and this seems out of scope for this review.
Lines 268 – 271: The wording of the sentence beginning “Interestingly, when exposed to a lethal dosage…” is confusing. Clarify by explaining that despite the different sensitivities between the parent and mutant strain to a low dose of light, at a certain point, a higher dose became lethal to both. Also, how are you defining a “dose” of light?
I also feel that the sentence beginning at line 271 is somewhat contradictory of the previous sentence, because based on number of genes, the work discussed in Reference #66 seems to highlight the importance of sigma B, whereas the work referenced in Reference #65 highlights the importance of other targets of light exposure.
Lines 286-301: What is ring-formation capacity? This is not a review aimed toward Listeria research, so I don’t think it’s fair to expect the reader to be familiar with this.
Lines 320-327. The paragraph beginning “Despite the apparent role of YtvA…” does not add any useful information and should be deleted.
Table 2: “Virulence factors”
Lines 342-343: Delete this sentence…future studies are also needed for many organisms. Why mention only B. cereus without making a specific case for it?
Two sections are labeled “3.1.3”
Line 563: The sentence regarding Tar and the ETC is vague. Expand the idea or delete the sentence.
Lines 582-583: S. agalactiae and E. faecium should not be mentioned here, as they are not susceptible to blue light therapy without the addition of photosensitizers
Line 589: What is rose bengal?
I find sections 5.1 – 5.5 to be completely unhelpful. These SOS summaries are too brief to be of any use. What was the rational in choosing the species included? S. enterica and Vibrio had not been mentioned in this review at all until this point – it feels entirely out of context.
Lines 747-752 do not offer information and should be deleted.
The authors at time refer to lethal vs. sub-lethal, but do not expand on this topic in terms of application and efficacy.
